# MnPc Films Deposited by Ultrasonic Spray Pyrolysis at Low Temperatures: Optical, Morphological and Structural Properties

**DOI:** 10.3390/ma16124357

**Published:** 2023-06-13

**Authors:** Anayantzi Luna Zempoalteca, José Álvaro David Hernández de la Luz, Adan Luna Flores, José Alberto Luna López, Alfredo Benítez Lara

**Affiliations:** 1Centro de Investigaciones en Dispositivos Semiconductores, Instituto de Ciencias, Benemérita Universidad Autónoma de Puebla, Prolongación 14 Sur, Colonia Jardines de San Manuel, Puebla 72570, Mexico; ana.lunazempoalteca@gmail.com (A.L.Z.); jose.luna@correo.buap.mx (J.A.L.L.); 2Facultad de Ingeniería Química, Benemérita Universidad Autónoma de Puebla, Avenida San Claudio y 18 Sur, Colonia Jardines de San Manuel, Puebla 72570, Mexico; adan.luna@correo.buap.mx; 3CONACYT—Centro de Investigaciones en Óptica, A.C, Loma del Bosque 115, Colonia Lomas del Campestre, León, Guanajuato 37150, Mexico; alfredbl@cio.mx

**Keywords:** manganese phthalocyanine (MnPc), ultrasonic spray pyrolysis, UV/Vis, Raman, XRD, SEM

## Abstract

In this work, we report how manganese phthalocyanine (MnPc) films obtained using the ultrasonic spray–pyrolysis technique at 40 °C deposited on glass substrate subjected to thermal annealing at 100 °C and 120 °C. The MnPc films were characterized using UV/Vis spectroscopy, Raman spectroscopy, X-Ray Diffraction (XRD), and Scanning Electron Microscopy (SEM). The absorption spectra of the MnPc films were studied in a wavelength range from 200 to 850 nm, where the characteristic bands of a metallic phthalocyanine known as B and Q bands were observed in this range of the spectrum. The optical energy band (Eg) was calculated using the Tauc equation. It was found that, for these MnPc films, the Eg has the values of 4.41, 4.46, and 3.58 eV corresponded to when they were deposited, annealing at 100 °C and 120 °C, respectively. The Raman spectra of the films showed the characteristic vibrational modes of the MnPc films. In the X-Ray diffractograms of these films, the characteristic diffraction peaks of a metallic phthalocyanine are observed, presenting a monoclinic phase. The SEM images of these films were studied in a cross-section obtaining thicknesses of 2 μm for the deposited film and 1.2 μm and 0.3 μm for the annealed films at 100 °C and 120 °C. Additionally, in the SEM images of these films, average particle sizes ranging from 4 to 0.041 µm were obtained. The results agree with those reported in the literature for MnPc films deposited by performing other techniques.

## 1. Introduction

Metal phthalocyanines (MPcs) belong to the group of organic compounds that are usually formed via carbon–carbon and carbon–hydrogen bonds [1,2,3,4,5]. Metal phthalocyanines are molecules that are studied in different fields for their physical and chemical properties [3,5,6,7,8] and their chemical inertness and thermal stability, which has led researchers to explore their technological utility in fields such as industrial dyes, catalysts, gas diffusion electrodes, organic photoconductors, photovoltaic solar cells, organic light-emitting devices, nonlinear optics, data storage optics, electro photography, near infrared absorbers, and photodynamic therapy, among others [5,6,7,8,9].

Metal phthalocyanine organic semiconductors exhibit interesting optical characteristics due to their ring structure [10]. The MPc shows two characteristic bands, with one of them called a B Band (known as a Soret Band) and the other one is called a Q Band (known as an α Band in porphyrin [10]). The latter is characterized by the π→π* transitions from HOMO (Higher Energy Molecular Orbitals Occupied by electrons) to LUMO (Lower Energy Molecular Orbitals unoccupied by electrons) [11,12] and, in the case of the B band, it is known by the deeper π→π* transitions in the UV/Vis spectrum [12]. Metal phthalocyanines are well known for their monoclinic structure according to their crystallographic records and from other various scientific reports [13,14].

Recently, metal phthalocyanines have been investigated for their application in the field of modern organic electronics due to their robust photo-physical and electro-physical properties [13,15,16,17]. Metal phthalocyanines such as zinc phthalocyanine (ZnPc), copper phthalocyanine (CuPc), tin phthalocyanine (SnPc), aluminum phthalocyanine, and vanadium phthalocyanine (OVPc) are, among others, the most studied phthalocyanines in the field of electronic organic devices [3,6,18]. While the properties of these phthalocyanines are relatively well known, there are very few studies of other MPcs. It has recently been pointed out that manganese phthalocyanine (MnPc) is very promising to be applied in the field of organic photovoltaics [6,9,18]. Due to the relatively high mobility of holes [7,19], it is possible that MnPc will also be found useful for application as hole-generating sources for the injection layer in organic light-emitting diode OLEDs. Therefore, a thorough investigation of its optical, morphological, and structural properties will be of great interest for modeling the performance of organic optoelectronic devices, as well as for the design of devices with efficient performance that can be achieved by modeling the charge transport in them.

The MPc can be deposited in the form of thin films. Thin films are layers of thin materials that grow from atoms or molecules that impinge on the surface of a substrate [20,21]. Thin films range in thickness from a few nanometers to a few hundred micrometers [20] and can be deposited by performing different techniques [22]. The latter can be divided into either chemical methods or physical methods, according to the nature of the deposit. The most commonly used methods include (a) Physical Vapor Deposition (PVD) methods that involve thermal evaporation deposition, pulsed laser deposition, molecular beam epitaxy, and sputtering; (b) Chemical Vapor Deposition (CVD) that involve organic metal CVD, low pressure CVD, atmospheric pressure CVD, and plasma enhanced CVD; and (c) Solution-Based Chemical (SBC) ones that involve chemical bath deposition, sol–gel, spin coating, dip coating, screen printing, and ultrasonic spray pyrolysis [22].

In chemical methods, precursor compounds are used, usually salts of some metallic element, and it is desired that these precursors react chemically on the substrate to be coated [22]. On the other hand, in physical methods, the material to be deposited is initially in powder form and, when some type of energy is applied to it, whether mechanical, electrochemical, or thermodynamic, the material is removed from the target and is then deposited on the substrate [22]. It is worth mentioning that physical methods require the use of low pressures through vacuum generation to ensure proper deposition and are therefore more expensive compared to chemical methods [22]. By contrast, chemical bath and ultrasonic spray–pyrolysis are the simplest and cheapest chemical techniques, due to their simplicity [22,23]. Ultrasonic spray–pyrolysis is an easy process to handle. It can be used to deposit over large areas, in addition to the fact that the deposit temperature is very low. It can be utilized for deposits on flexible materials; in this process, it is possible to use a wide variety of precursors, and the cost of this process is inexpensive compared to other techniques. Additionally, it does not include the control of high-pressure processes. This process has the capability of growing multilayer structures and can have the freedom to calibrate deposition variables such as deposition temperature, deposition substrate, precursor composition and concentration, and solvent composition. Pyrolysis (from the Greek pyro (fire) and lysis (break)) is one of the types of reaction that is induced to obtain the deposit of a certain desired material. The energy to carry out the reaction is provided by the increase in temperature; therefore, the word thermolysis is an equally applicable term. As its etymology suggests, applying heat to the precursor material to break its chemical bonds results in the formation of a new compound [24,25].

Figure 1 shows the mechanism for the formation of solid particles via spray pyrolysis from droplets. Initially, the droplets are obtained by using an atomizer that evaporates to transform into a gas phase via ultrasonic. The precursor to the liquid molecular species vibrates and produces a mist or vapor spray. The decomposition consists of several reactions called precursor pyrolysis that depend on the nature of the precursor fraction and the type of solvents used. Both the material and the solvent exhibit a saturation energy. Therefore, the obtained gaseous values are strongly supersaturated and stochastically firm collisions to enable the formation and deformation of molecular clusters, which depend on the nucleation (homogeneous and heterogeneous) and process parameters to achieve aggregation, coalescence, shallow growth, and crowding [26].

When the pyrolysis spray is deposited, you will see ions that are formed due to the vibration that exists in the atomizer with the material where the material will react molecularly, allowing its band gap to vary due to the presence of oxygen vacancies in the material [27,28].

In this context, we report obtaining the MnPc films that were deposited via ultrasonic Spray–Pyrolysis at low temperatures. These films were analyzed using different characterization techniques (UV/Vis, Raman, XRD, and SEM) to know their optical, morphological, and structural properties. It is expected that these films may be applied in the field of optoelectronic devices.

## 2. Experimental Work

### 2.1. Synthesis of MnPc Films

The manganese phthalocyanine II was purchased from Sigma Aldrich at 90% purity and was used as solute for the preparation of the solution with ethanol (J.T.Baker) as a solvent. The reagents were used without any further purification.

A Baku BK 2000 ultrasonic bathtub was used to enhance the homogeneous mixing of the solution in the form of suspension. The MnPc was a precursor solution; it is atomized, and the molecules vibrate, breaking the bonds between them, transitioning from a liquid state to a vapor state, forming solid particles in the substrate with a specific temperature [24,25].

Corning glasses were used as substrates for depositing the MnPc films, and a thermal grill (Chemat Technology, model TW-4H) was used for thermal annealing of the MnPc films.

To obtain the manganese phthalocyanine II (MnPc II) films, manganese phthalocyanine II and ethanol were used, and a solution of MnPc and ethanol with a concentration of 0.13 g/mL was made. The solution was placed in the ultrasonic system at 50 °C. The films were deposited on a thermal grill for 30 min on a conventional Corning glass substrate at 40 °C deposition temperature (TD) (sample labeled as A1). The films were then thermally annealed on the thermal grill to remove any impurities at a temperature (Tr) considering two values: at 100 °C for 1 h (A21) and at 120 °C for 2 h (A22). After the films were obtained, they were kept at room temperature. The MnPc films were measured for from one to three months and had no changes.

### 2.2. Characterization of the Films

A Varian (Agilent) Cary 5000 wide range UV-VIS-NIR system (with a PbS mart detector, with a measuring range from 175 nm to 3300 nm) was used to characterize the grown films. The absorption spectra were measured at a normal incidence in a spectral range from 200 to 850 nm. The Raman spectra were measured in a range from 400 to 1800 cm^−1^ on a Horiba-JOBIN YVON MicroRaman system, model LabRAM-HR, with a Helium-Neon laser at 632.8 nm. The diffractograms of the MnPc films were measured using an X-Ray diffractometer, model D2 Phaser, Bruker brand, with a Bragg–Brentane geometry, Cu 1.54184 tube [Aº], LYNXEYE detector. The measuring range of 2*θ* is from 0 to 60° and the sample measurement was horizontal. The SEM images of the MnPc films were measured using an Ultra-High Resolution Scanning Electron Microscope JSM-7800F Schottky Field Emission Scanning Electron Microscope (JEOL). Cross-sectional measurements were performed for the MnPc films and μm-scale images were obtained.

## 3. Results and Discussion

Figure 2a shows the UV-Vis spectrum of the MnPcs films. As can be seen, the absorbance spectra of the samples A1 and A21 are stronger than that of the A22 film; additionally, they exhibit the characteristic B band around 261, 263, and 224 nm. A large decrease in the strength of the B-band absorption peak of the A22 sample located around 224 nm can be observed, which is due to the orbital overlap of the phthalocyanine ring and the central metal ion [11,12]. On the other hand, the three samples exhibit the Q band, which lies at 740 nm and shows a weak intensity for the three films [11,12]. Another band is presented at 352 nm, which is a characteristic band of the MnPc. It is known that the ultraviolet-visible (UV-vis) absorption spectra of metallic phthalocyanines are the result of their widely conjugated π electrons and the overlapping orbitals of the central metal ion. The B-band is associated with the central metal arising from the π-d [29] transitions.

The optical band gap (*Eg*) of the samples was determined from the UV/Vis spectra using the Tauc [10] equation (Equation (1)):(1)$${\left(\alpha h\nu \right)}^{\frac{1}{n}}=C_{1}\left(h\nu -E_{g}\right)$$
where *α* = Absorption coefficient, *h* = Planck’s const., *v* = incident radiation frequency, $C_{1}$= proportionality constant, *E_g_* = bandgap energy of the material, and *n* = power whose value depends on the electronic transition presented by the material (MnPc presents a direct allowed transition for which *n* = 0.5 [10]).

The Tauc extrapolation diagram for the abscissa provides the value of the optical band gap (*Eg*) [10]. Figure 1b shows an example to obtain the forbidden band of the MnPc films; in this case, it corresponds to the A21 sample for which *Eg =* 4.4461 eV. The obtained remaining values of the MnPc films are as follows: *Eg =* 4.41 eV (A1 sample) and *Eg =* 3.58 eV (A22 sample); we point out that all these values are very close to those reported in the literature [10,30,31,32,33].

On the other hand, Figure 3 displays the Raman spectra of the MnPn films. Considering this information, we generate Table 1, which lists the wavenumbers and vibrational modes found in the Raman spectra. Each vibrational mode is described according to its molecule type, and such a frequency vibrational mode is like that reported in the literature. The spectra (Figure 3) show, via characteristic vibrational frequencies, the presence of the rings or benzenes of the metal phthalocyanine in the ranges of 677 cm^−1^ (benzene strain), 1593 cm^−1^ (benzene, C_δ_H strain, in-plane bending), and 1610 cm^−1^ (benzene ring strain). Ring–metal bonding is observed in the ranges 591 cm^−1^ (ring, NM, benzene ring strain, elongation), 751 cm^−1^ (CN_m_CNM, pyrrole strain, elongation), 950–1102 cm^−1^ (NM, CNmC-in-plane bending, Isoindol strain, CH-in-plane bending, Isoindol NM strain), 1190–1394 cm^−1^ (C_δ_H-in-plane bending, pyrrole NM strain, isoindol NM strain, CH-in-plane bending, CH CNmC-in-plane bending, pyrrole CδCδNM strain, CNmCCδH-in-plane bending), 1518 cm^−1^ (CN_m_C strain, pyrrole elongation, CH-in-plane bending); molecules around the inner ring and externally occur in the ranges of 831 cm^−1^ (CH-out-of-plane bending), 1133 cm^−1^ (CH-in-plane bending), and 1430 cm^−1^ (CH-in-plane bending, C_β_C_β_ strain) [34]. Variations in the intensities of the vibrational modes show that the annealing process realized on the MnPc films causes a reordering in the molecular structure on the films as a consequence of the thermal energy absorption.

The X-Ray diffraction patterns of the MnPc films are shown in Figure 4. The diffractograms exhibit the characteristic diffraction peaks for the monoclinic phase of the MnPc films; this characteristic phase is confirmed by both the crystallographic cards and the reports found in the literature [18,35,36,37,38,39,40,41]. The monoclinic phase is found in the vast majority of metallic phthalocyanines where it is considered a dominant phase [42,43]. All diffraction peaks present the same phase, as listed in Table 2. An outstanding characteristic observed in the diffraction patterns is that the peaks are broad and tend to exhibit an amorphous behavior. The annealing process applied to the MnPc films modified the diffraction peaks as it is observed in the XRD diffraction patterns. The films A1 and A22 show a great majority of diffraction peaks compared to film A21; this is due to the temperature effects where the molecules seek their chemical stability.

On the other hand, the morphology of the MnPc films was determined via the SEM images. Figure 5a,c,e show the cross-sectional SEM images for each film; from these images, the thickness of such films as 2.009 µm, 1.232 µm, and 0.374 µm for samples A1, A21, and A22, respectively, are estimated. The films A1, A21, and A22 present small lumps/clumps of particles on the surface of the film where the thicknesses decrease with the presence of the temperature. When the molecules are vibrating, they will break bonds and seek their molecular order or chemical stability [44,45].

The films obtained via ultrasonic spray pyrolysis have a uniform thickness and it is possible to have good control of their thickness, which allows them to be used in different applications.

Additionally, Figure 5b,d,f show the surface morphology of the MnPc films. The average particle sizes for samples A1, A21, and A22 are 0.169 µm, 1.77 µm, and 2.204 µm, respectively. The SEM images show circular or spherical particles and some of them accumulate to form aggregates with several spherical particles, forming larger clusters. The size of the average particles grows due to the presence of small crystals, where the molecules seek their molecular order when witnessing the temperature as shown in the XRD diffractograms of samples A1, A21, and A22 where their band gap decreases due to the presence of these crystals.

It is known that the presence of temperature produces different oxygen species when the material is transformed. Given that oxygen species play a crucial role in the formation of monofaces, it therefore needs heat treatment, that is, calcinations at different temperatures for its crystallization until it achieves a pure polycrystalline character. In other words, the phase formation energy requires temperature optimization to achieve a single phase of the compound [46,47].

The presence of oxygen or oxygen vacancies causes the band gap of the material to vary as the temperature increases and that it presents different phases in the MnPc films, varying the size of the particles of each film [46,47].

In film A1, there is a greater presence of oxygen vacancies on the surface of the film; however, when applying temperature, these oxygen presences or vacancies will be calcined to form monofaces in films A21 and A22, where the value of the band gap and the size of the particles that will be dispersed on the surface of films A21 and A22 will change.

Studies on the optical, morphological, and structural properties of MnPc films have been previously conducted by K. R. Rajesh [13], B.R. Sridevi [48], and Dapeng Li [14], with the purpose of being applied in a specific area. They used different forms of deposit to know their physical and chemical properties.

## 4. Conclusions

The MnPc films were obtained from easily handled precursors at a low temperature of 40 °C (A1), which, after annealing at 100 °C (A21) and 120 °C (A22), their optical, structural, and compositional properties were modified, as well as the films’ present thicknesses of 2.009 µm, 1.232 µm, and 0.374 µm. In the UV/Vis spectra, the characteristic bands Q and B were obtained at the positions of 740 nm and 263 nm, respectively. From the Raman spectra, the characteristic vibrational modes of an MnPc and its characteristic molecule type, which was identified according to their vibrational modes, were obtained. The XRD diffractograms confirm the crystallographic structure of the MnPc film together with its characteristic planes of structures that result in the monoclinic phase. The SEM images showed the morphological characteristics of small clusters dispersed on a uniform film deposited below these clusters or particles; the average particle size was obtained for samples A1, A21, and A22, obtaining the values of 2.319 µm, 0.041µm, and 4.465 µm, respectively.

## Figures and Tables

**Figure 1 materials-16-04357-f001:**
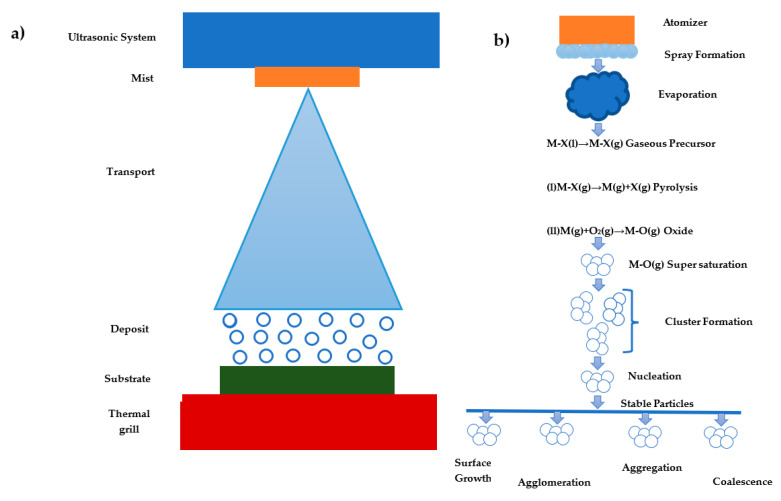
Pyrolysis spray formation process: (**a**) physical appearance and (**b**) detailed chemical appearance [26].

**Figure 2 materials-16-04357-f002:**
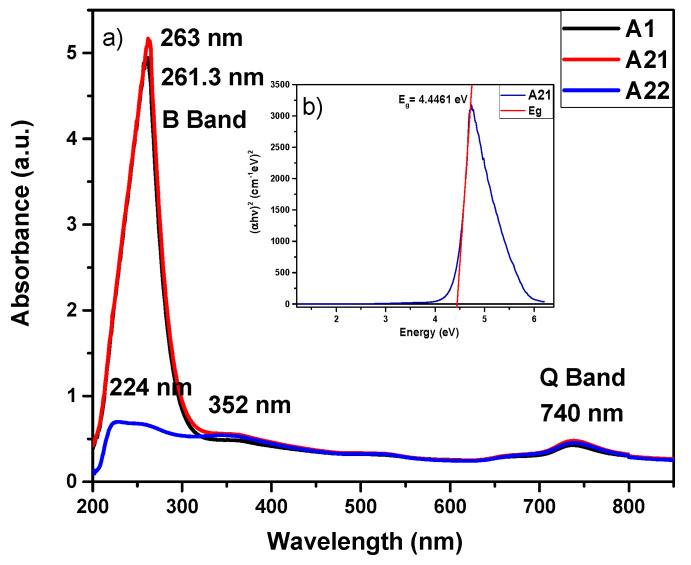
(**a**) Absorbance spectra of the MnPc films for samples A1, A21, and A22 and (**b**) determination of the forbidden energy band of the MnPc film (A21) obtained using the Tauc equation.

**Figure 3 materials-16-04357-f003:**
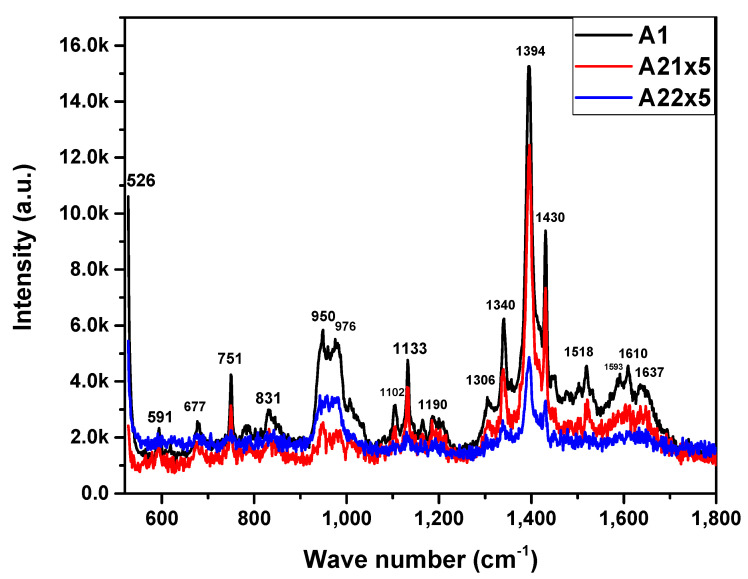
Raman spectra of MnPc films for samples A1, A21, and A22.

**Figure 4 materials-16-04357-f004:**
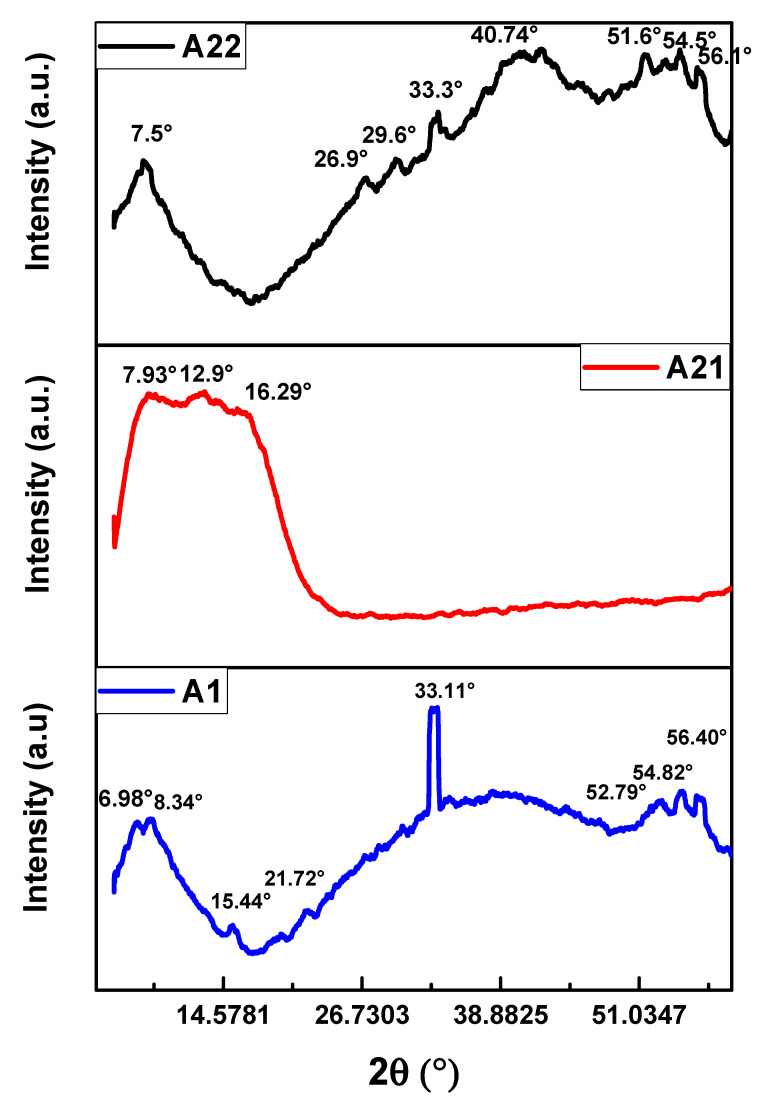
Diffractograms of the MnPc films for samples A1, A21, and A22.

**Figure 5 materials-16-04357-f005:**
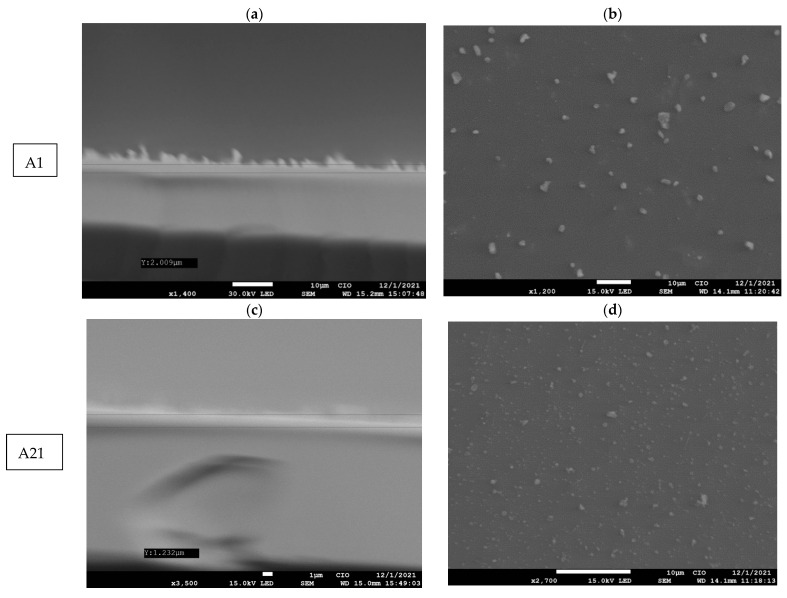
(**a**,**c**,**e**) SEM images (cross section) of MnPc films A1, A21, and A22, and (**b**,**d**,**f**) SEM images of the surface of the MnPc films A1, A21, and A22.

**Table 1 materials-16-04357-t001:** Descriptive table of Raman spectra of MnPc films [5,6,7,8,9,10].

v (cm^−1^)	Description	Vibration Modes
526	Si Substrate	
591	Ring NM Benzene	Stretching in the ring Stretching Elongation
677	Benzene	Strain
751	CN_m,_CNM Pyrrole	Tension Elongation
831	CH	Out-of-plane bending
950	NM, CN_m_C Isoindole	In-plane bending Deformation
976	NMCN_m_C Isoindole	In-plane bending Deformation
1102	CH Isoindole NM	In-plane bending Tension
1133	CH	In-plane bending
1190	C_ẟ_H Pyrrole, NM	In-plane bending Tension
1306	Isoindole NM CH	Tension In-plane bending
1340	Isoindole NM CH CN_m_C	Tension In-plane bending
1394	Pyrrole C_ẟ_C_ẟ_NM CN_m_CC_ẟ_H	Tension In-plane bending
1430	CH C_β_C_β_	In-plane bending Tension
1518	CN_m_C Pyrrole CH	Tension Elongation In-plane bending
1593	Benzene C_ẟ_H	Tension In-plane bending
1610	Benzene	Ring tension

**Table 2 materials-16-04357-t002:** Crystallographic planes and crystal structure shown in the diffractogram [6,9,18,19,20,21,25,26].

2Ɵ (°)	h	k	l	Crystalline System
A1 Film				
8.34	1	0	**1**	Monoclinic
15.44	1	0	**2**	
21.72	1	1	**3**	
33.11	2	2	**0**	
52.79	4	2	**2**	
54.82	5	1	**1**	
56.40	5	1	**1**	
A21 Film				
7.93	0	2	**0**	Monoclinic
12.9	2	1	**0**	
16.29	1	0	**2**	
A22 Film				
7.5	0	1	**1**	Monoclinic
26.9	−3	1	**2**	
29.6	−3	1	**5**	
33.3	2	2	**0**	
40.74	4	0	**0**	
51.6	4	2	**2**	
54.5	4	2	**2**	
56.1	5	1	**1**	

## Data Availability

Not applicable.

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
