# Peer review of "MnPc Films Deposited by Ultrasonic Spray Pyrolysis at Low Temperatures: Optical, Morphological and Structural Properties"

_materials, 2023, doi:10.3390/ma16124357_

Round 1
Reviewer 1 Report
The manuscript by Anayantzi Luna Zempoalteca1, J. A. David Hernández de la Luz, Adan Luna Flores, José Alberto Luna López and Alfredo Benítez Lara presents preparation and structural characteristic (UV/Vis and Raman spectroscopy, X-ray "powder" diffraction and scanning electron microscopy) of manganese phthalocyanine (MnPc) films obtained by ultrasonic spray-pyrolysis technique.
Metal phthalocyanines (MPcs) enable high potential for practical applications in novel molecular electronic and optical devices.
MPcs are used as organic photoconductors, photovoltaic solar cells and in photodynamic diagnosis and photodynamic therapy.
The reviewed work, submitted to section: Advanced Materials Characterization, is worth publishing but requires major revision.
Questions and recommendations:
1) Is the monoclinic form of the tested compound the only one?
2) Have phase transitions been observed for manganese phthalocyanine (MnPc)?
3) Is the MnPc stable at room temperature or below 0°C?
4) Table 2 - 2? values should be given in degrees.
5) Table 2 - what does it mean "crystal structure of each peak"?
6) Could you check references 14 and 16.
Author Response
Hello Reviever 1
I am sending you the answers to the questions that you sent me about the article.
After the references, follow the questions with their respective answers
I wish you an excellent day and greetings.

Reviewer 2 Report
The authors in the present work show MnPc films deposited by ultrasonic spray pyrolysis at low 1 temperatures: optical, morphological and structural properties. Manuscript written in well manner, systematic and work is also interesting. But in a Scientific and Fundamental point of view for global readers, some points must be clarified.
· Why do authors only consider only MnPc films why not others? In what point of view/ application do authors fabricate these films? (in the application or low-temperature synthesis)
· What is the role of ions in spry pyrolysis must be explain in introduction by referring https://doi.org/10.1039/D0CC06922A and https://doi.org/10.1039/D1SC03775G
· Since absorption is there authors can calculate the bandgap and show what deviation of these films w. r. t. temperatures. How phases and bandgap is varies must be explain in bried by reefing https://doi.org/10.1002/adom.202000901 and https://doi.org/10.1039/D2TC04310F
· Raman Spectra must be labelled w.r.t bands.
· Authors must rewrite fig 3 captions fully. In A1 films why 33.11 peak show-wide?
· In X-ray diffractograms if all the films belong to monoclinic what diffraction is different for all three films?
· Overall novelty is not seen why authors need to work on this. On what parameters it makes differ from other prepared films?
· Thickness of films must be include.
English must be review once more time.
Author Response
Hello Reviewer 2
I am sending you the answers to the questions you sent me about the article.
After the references, it follow the questions with their respective answers.
I wish you an excellent day and greetings.

Round 2
Reviewer 2 Report
Still author does not include the references as per my previous comments.
Upon inclusion of these comments we may accept this paper.
Author Response
Hello Reviewer 2
I sent you the answers to the questions than I couldn´t answer well.
I appreaciate your comments and suggestions to improve the article for the material magazine.
